# Life expectancy and active life expectancy by disability status in older U.S. adults

**Haomiao Jia** [1]*, **Erica I. Lubetkin** [2]

**1** Department of Biostatistics, Mailman School of Public Health and School of Nursing, Columbia University, New York, New York, United States of America, **2** Department of Community Health and Social Medicine, City University of New York School of Medicine, New York, New York, United States of America

* hj2198@columbia.edu

**Data Availability Statement:** Data cannot be shared publicly because this study used the Health Outcomes Survey Limited Data Set (LDS). The dataset contains potentially identifying or sensitive patient information (e.g., participants' zip code,

## Abstract

### Objectives

The Medicare Health Outcome Survey (HOS) is the largest longitudinal survey of the U.S. community-dwelling elderly population. This study estimated total life expectancy, active life expectancy (ALE), and disability-free life expectancy (DFLE) by disability status among HOS participants.

### Methods

Data were from the Medicare HOS Cohort 15 (baseline 2012, follow-up 2014). We included respondents aged ≥ 65 years (n = 164,597). Participants' disability status was assessed based on the following six activities of daily living (ADL): bathing, dressing, eating, getting in or out of chairs, walking, and using the toilet. The multi-state models were used to estimate life expectancy, ALE, and DFLE by participants' baseline disability status and age.

### Results

Persons who had higher-level ADL limitations had a shorter life expectancy, ALE, and DFLE. Also persons with disability had greater expected life years with disability than those with no limitations and those with mild limitations. For example, among 65-year old respondents with no limitations, mild limitations, and disability, life expectancy was 19.9, 18.6, and 17.1 years, respectively; ALE was 14.0, 9.5, and 7.2 years, respectively; DFLE was 17.3, 15.2, and 11.4 years, respectively; and expected years with disability was 2.6, 3.4, and 5.7 years, respectively.

### Conclusions

This study demonstrated that greater levels of disability adversely impact life expectancy, ALE, DFLE, and expected number of years with a disability among U.S. older adults. Understanding levels of disability, and how these may change over time, would enhance health care quality and planning services related to home care and housing in this community-dwelling population.

date of birth, date of death, etc.). A signed Data Use Agreement (DUA) with CMS is required to obtain LDS data files. The data file was obtained through a DUA between the U.S. Center for Disease Control and Prevention (CDC) and the CMS. In this DUA, the corresponding author (HJ) is one of custodians of the data file. The authors do not have the authority share the data publically. The information about requiring for the HOS LDS is available at https://www.hosonline.org/en/data-dissemination/research-data-files/. The request for the HOS LDS files should be submitted to the CMS Information and Technical Support at hos@hsag.com.

**Funding:** The authors received no financial support for the research, authorship, and/or publication of this article.

**Competing interests:** The authors declared no potential conflicts of interest with respect to the research, authorship, and/or publication of this article.

## Introduction

The United States population has experienced a shift in its age structure, with the number of persons aged 65 and older projected to nearly double between 2012 and 2050 [1]. While the life expectancy for the U.S. population has increased over time, a higher proportion of the elderly population is living with chronic diseases, activity limitations, and disability, after accounting for age differences [2–4]. However, what remains unclear is whether longer life expectancy is associated with the delayed onset of morbidity/disability and increases in recovery, or more years living with morbidity/disability through keeping individuals with chronic comorbid conditions alive [5, 6].

Life expectancy and years of life living with disability are affected by many factors, including incidence and recovery rates of disability as well as disability-associated mortality risk [7, 8]. Because of this, quantifying the impact of disability on life expectancy and years living with a disability would be integral and consistent with the overarching goal of Healthy People 2030 to "attain healthy, thriving lives and well-being, free of preventable disease, disability, injury and premature death" [9]. To date, disability-adjusted life expectancy (DFLE) or active life expectancy (ALE), defined as expected remaining life years spent in a non-disabled or "active" state, have commonly been calculated [10, 11]. These two terms (DFLE and ALE) are used interchangeably in the literature and provide a way to assess the disability-associated burden of disease [12–15].

Knowing an individual's total number of expected remaining life years and expected remaining life years in a non-disabled or "active" state based on this person's age and disability status is particularly useful for both patients and health care providers in order to facilitate measuring health care quality and planning related services [16]. However, the ordinary life table method cannot provide such estimates because a person's disability status may change during his/her lifetime [6, 17]. For example, a person may be healthy and free from any disability at younger ages and then, in later life, develop one or more chronic illness and, therefore, became disabled. Similarly, a disabled person may recover from illness or injury and be able to perform activities of daily living (ADLs) without assistance. To solve this problem, one can use multi-state models to analyze complex transitions among multiple states [13, 18, 19]. The multi-state model treats disability as a temporary transitional state rather than as an irreversible state and allows transfers from one state to another during the remaining lifetime. This method often has been used to estimate the probability and average length of time that persons who began with a specific state will be in another state [20]. Therefore, this method can be used to estimate life expectancy as well as ALE and DFLE by participants' disability status [19, 21–23].

One of the biggest weaknesses of the multi-state modeling method is the high data requirement, as longitudinal data are needed to estimate probabilities of transferring from one state to another state. This weakness has limited the use of this method to provide such estimates for the U.S. general population [15, 24]. The Medicare Health Outcome Survey (HOS) is the largest longitudinal survey of the United States community-dwelling elderly population [25]. The present study estimates life expectancy, as well as DFLE and ALE, by baseline disability status among HOS participants. In this study, we used a binary disability measure (presence of activity limitation), a 3-level disability measure (no limitation, mild limitation, and disability), and a 5-level disability measure (no, mild, moderate, severe, and complete limitation).

## Materials and methods

The data for this study were obtained from a Medicare Health Outcomes Survey (HOS) limited data set available through the U.S. Centers for Medicare & Medicaid Services

(CMS). The study was reviewed and approved by the Columbia University Medical Center Institutional Review Board. The Medicare HOS is a nationwide annual survey of Medicare beneficiaries. Each year, the HOS randomly selects a cohort of Medicare beneficiaries who voluntarily enrolled in Medicare Advantage private health plans [25]. The selected individuals who complete a baseline survey are resurveyed two years later. This study used the HOS Cohort 15 whose baseline data were collected in 2012 and follow-up data were collected in 2014. The date of death is available if death occurred by January 31, 2015. We included all respondents who were aged 65 years or older and alive at the baseline survey and participated in the baseline survey. The total sample for this study was 164,597. Among the sample, 100,290 (61%) were alive at the follow-up survey and completed the follow-up survey, 26,111 (16%) died before the follow-up survey, and 38,196 (23%) did not participate in the follow-up survey. An additional 88 participants died after completing the follow-up survey.

At both baseline and follow-up surveys, the HOS asks respondents whether they have difficulty with the following six basic ADLs: (1) bathing, (2) dressing, (3) eating, (4) getting in or out of chairs, (5) walking, and (6) using the toilet [26]. These questions have been used for the classification of respondents' disability status [8, 11–25, 27]. In this study, we used the ADL staging method developed by Stineman and colleagues (2014) to measure disability [27]. This method classifies respondents into the following five ADL/disability stages:

- Stage 0: no difficulty for all six activities

- Stage I: mild limitation (eating, toileting, dressing, and bathing are not difficult; have difficulty with getting in or out of chairs and/or walking)

- Stage II: moderate limitation (eating and toileting are not difficult; have difficulty with bathing and/or dressing)

- Stage III: severe limitation (have difficulties with eating and/or toileting, but not all six activities)

- Stage IV: complete limitation (have difficulties with all six activities).

We used a binary measure as having **activity limitation** if a respondent reported having a limitation for any of these six ADLs (stages I-IV vs. stage 0). We also used a 3-level disability measure by dividing the "activity limitation" state described above into two exclusive states: **mild limitation** (stage I) and **disabled** (stages II-IV). In this study, we classified those who had no limitation or had a mild limitation as **non-disabled** (stages 0-I). Finally, we examined a 5-level disability measure (stage 0 to stage IV).

Previous studies used slightly different disability measures to calculate ALE and DFLE [11–15, 24]. In this study, we defined and calculated ALE as expected remaining life years with no activity limitation (stage 0) and DFLE as expected years of life remaining in a non-disabled state (stages 0 and I) [12].

## Statistical analysis

Multi-state models were used to estimate average number of total remaining life years at a given age (i.e., life expectancy) and number of remaining years of life in a non-disabled or active state at a given age (i.e., DFLE or ALE) for cohorts of persons by their baseline disability status and age [18, 29]. Because the HOS data were collected at baseline and at follow-up after 2 years, we estimated life expectancy and ALE/DFLE at ages 65, 67, . . ., and 95 years. To illustrate this method, we describe a Markov process with $k$ transient states $s = (1,2,. . .,k)$ for $k$

levels of disability measure and one absorbing state $s = k+1$ for dead. Let $p_t^{i,j} = \Pr(s_{t+2} = j | s_t = i)$ be transition probability from state $i$ at age $t$ to state $j$ at age $t+2$.

Because time intervals between baseline and follow-up varied from person to person, we estimated the instantaneous transition rates between different disability states, $\mu_t^{i,j} = \lim_{\Delta \to 0} \frac{\Pr(s_{t+\Delta}=j|s_t=i)}{\Delta}$, from log-linear models with age as a time-dependent predictor [18, 19]. We obtained transition probabilities between different disability states, $p_t^{i,j} = 1 - exp(-2\mu_t^{i,j})$, assuming a constant instantaneous transition rate in the age interval [19, 28, 29]. The probability of death for each disability state during each age interval was estimated based on the probability of death for the total population and hazard ratio of death for each disability state relative to the reference group (non-disabled) at different ages. We used the probability of death from the 2012 U.S. life tables as the probability of death for the total population and estimated hazard ratios using a Cox proportional hazard model with time-varying covariates from the HOS data [2, 30].

For an age cohort of individuals that the numbers of persons in each states $i$ at the starting age $x$, $l_x^i$, are known, the expected numbers of persons in each states at ages $x+2$, $x+4$,..., can be obtained iteratively based on transition probabilities as $l_{t+2}^i = l_t^i(1 - \sum_{j=1, j\neq i}^{k+1} p_t^{i,j}) + \sum_{j=1, j\neq i}^{k} l_t^j p_t^{j,i}$, $(i = 1, 2, \ldots, k)$. Let $L_t^i$ be number of years lived in state $i$ during the age interval from $t$ to $t+2$ for the age cohort. We estimated $L_t^i$ using the trapezoidal rule [2, 19, 29]. The expected number of remaining life years in state $i$ for this age cohort is $e_x^i = (\sum_{t \geq x} L_t^i)/l_x$ where $l_x = \sum_{i=1}^{k} l_x^i$ is the total number of persons at the starting age $x$. Suppose state $s = 1$ is the non-disabled (or "active") state, the expected years of life remaining in state $s = 1$, $e_x^1$, is DFLE or ALE for this age cohort. The total life expectancy for this age cohort is $e_x = \sum_{i=1}^{k} e_x^i$.

Observations with missing value in disability status (about 2% at baseline and 6% at follow-up) were excluded from estimating transition probabilities between different disability states. We used the bootstrap method to estimate the standard error of the estimated life expectancy and ALE [19].

## Results

At baseline, the average participant age was 75.1 years; 53% of participants were between 65 and 74 years old, 34% were between 75 and 84 years old, and 13% were 85 years or older (Table 1). Women comprised 58% of the sample, and white non-Hispanics constituted 76% of the sample. About 62% of participants reported no limitation, 16% reported mild limitations, 9% reported moderate limitations, 8% reported severe limitations, and 4% reported complete limitations. At follow-up, 67% of participants reported no limitation, 18% reported mild limitations, 7% reported moderate limitations, 6% reported severe limitations, and 2% reported complete limitations.

### Using a binary disability measure

For a binary disability measure (presence or absence of activity limitation), Table 2 presents total years of life remaining (i.e., life expectancy, $e_x$), years of life remaining in no limitation state (i.e., ALE, $e_x^1$), and years of life remaining in an activity limitation state ($e_x^2$) for the total sample, those who did not have an activity limitation, and those who had an activity limitation, at different ages, respectively. For example, a 65-year person was expected to live an additional 19.3 years. Of these 19.3 years, 12.5 years (65%) were without a limitation, and 6.7 years (35%) were in an activity limitation state.

Persons who did not have an activity limitation had a longer life expectancy and a longer ALE than persons who had an activity limitation of the same age. Also, persons who did not have an activity limitation were expected to spend a higher percentage of their remaining life

**Table 1. Sample characteristics at the baseline and the follow-up.**

| | Baseline N = 164,597 | | Follow-up N = 100,290 | |
|---|---|---|---|---|
| | **N** | **Percent** | **N** | **Percent** |
| Age, Mean (SD) | 75.1 (7.4) | | 76.2 (6.7) | |
| 65–74 | 87,972 | 53% | 47,929 | 48% |
| 75–84 | 55,676 | 34% | 39,337 | 39% |
| 85–94 | 19,313 | 12% | 12,308 | 12% |
| 95+ | 1,636 | 1% | 716 | 1% |
| Female | 95,115 | 58% | 58,519 | 58% |
| Race/ethnicity | | | | |
| White non-Hispanics | 121,334 | 76% | 77,694 | 78% |
| Black non-Hispanics | 13,031 | 8% | 7,427 | 7% |
| Hispanics | 15,735 | 10% | 8,803 | 9% |
| Other | 9,404 | 6% | 5,408 | 5% |
| Disability status | | | | |
| No limitation (Stage 0) | 100,475 | 62% | 62,680 | 67% |
| Mild limitation (Stage I) | 26,418 | 16% | 16,650 | 18% |
| Moderate limitation (Stage II) | 14,613 | 9% | 6,861 | 7% |
| Severe limitation (Stage III) | 13,095 | 8% | 5,763 | 6% |
| Complete limitation (stage IV) | 6,397 | 4% | 2,024 | 2% |

**Table 2. Total expected life years and expected life years living with and without activity limitation overall and by each of two initial disability states for U.S. older adults.**

| Age ($x$) | Total sample | | | Initial disability status at age $x$ | | | | | | | |
|---|---|---|---|---|---|---|---|---|---|---|---|
| | | | | No limitation | | | Activity limitation | | | Difference[d] | |
| | $e_x$ [a] | $e_x^1$ [b] | $e_x^2$ [c] | $e_x$ | $e_x^1$ | $e_x^2$ | $e_x$ | $e_x^1$ | $e_x^2$ | total | active |
| 65 | 19.3 | 12.5 | 6.7 | 19.9 | 14.0 | 5.9 | 17.8 | 8.9 | 8.9 | 2.1 | 5.1 |
| 67 | 17.8 | 11.4 | 6.4 | 18.3 | 12.7 | 5.6 | 16.2 | 7.8 | 8.4 | 2.1 | 4.9 |
| 69 | 16.3 | 10.3 | 6.0 | 16.9 | 11.6 | 5.3 | 14.8 | 6.8 | 8.0 | 2.1 | 4.8 |
| 71 | 14.9 | 9.2 | 5.7 | 15.4 | 10.4 | 5.0 | 13.3 | 5.8 | 7.6 | 2.1 | 4.7 |
| 73 | 13.5 | 8.1 | 5.4 | 14.1 | 9.4 | 4.7 | 12.0 | 4.8 | 7.1 | 2.1 | 4.5 |
| 75 | 12.2 | 7.1 | 5.1 | 12.8 | 8.4 | 4.4 | 10.7 | 4.0 | 6.7 | 2.1 | 4.4 |
| 77 | 10.9 | 6.1 | 4.8 | 11.6 | 7.4 | 4.1 | 9.5 | 3.2 | 6.3 | 2.0 | 4.2 |
| 79 | 9.7 | 5.2 | 4.5 | 10.4 | 6.6 | 3.8 | 8.4 | 2.6 | 5.8 | 2.0 | 4.0 |
| 81 | 8.6 | 4.4 | 4.2 | 9.3 | 5.8 | 3.5 | 7.4 | 2.0 | 5.4 | 1.9 | 3.8 |
| 83 | 7.5 | 3.7 | 3.9 | 8.3 | 5.1 | 3.2 | 6.5 | 1.6 | 4.9 | 1.8 | 3.5 |
| 85 | 6.6 | 3.0 | 3.6 | 7.3 | 4.5 | 2.9 | 5.6 | 1.2 | 4.5 | 1.7 | 3.3 |
| 87 | 5.7 | 2.4 | 3.3 | 6.5 | 3.9 | 2.6 | 4.9 | 0.9 | 4.0 | 1.6 | 3.0 |
| 89 | 5.0 | 2.0 | 3.0 | 5.7 | 3.4 | 2.3 | 4.3 | 0.7 | 3.6 | 1.5 | 2.8 |
| 91 | 4.3 | 1.6 | 2.7 | 5.1 | 3.0 | 2.0 | 3.7 | 0.5 | 3.2 | 1.3 | 2.5 |
| 93 | 3.7 | 1.3 | 2.5 | 4.5 | 2.7 | 1.8 | 3.2 | 0.4 | 2.9 | 1.2 | 2.3 |
| 95 | 3.2 | 1.0 | 2.2 | 3.9 | 2.4 | 1.6 | 2.8 | 0.3 | 2.6 | 1.1 | 2.1 |

[a]: Total remaining life years (i.e., life expectancy) for persons of age $x$.

[b]: Remaining life years with no limitation (i.e., active life expectancy) for persons of age $x$.

[c]: Remaining life years with activity limitation for persons of age $x$.

[d]: difference in total life expectancy (total) and active life expectancy (active) between those without and with activity limitation; all differences are significantly different from 0 ($p < 0.0001$). Standard errors of estimates are available in S1 Table.

years without a limitation. For example, 65-year old persons without an activity limitation had a life expectancy of 19.9 years and ALE of 14.0 years. By contrast, 65-year old persons with an activity limitation had a life expectancy of 17.8 years and ALE of 8.9 years. Therefore, having an activity limitation at age 65 was associated with a 2.1-year (10%) decrease in life expectancy and a 5.1-year (36%) decrease in ALE. All estimates are reliable with small standard errors (S1 Table). When examined according to gender, similar results were observed for both men and women (Table 3).

## Using a 3-level disability measure

For a 3-level disability measure (no limitation, mild limitation, and disabled), Table 4 presents total life expectancy ($e_x$), ALE ($e_x^1$), life expectancy in a mild limitation state ($e_x^2$), and life expectancy in a disability state ($e_x^3$), respectively, for the total sample, and for participants by the 3 initial disability state. The sum of $e_x^1$ and $e_x^2$ is expected life years remaining in a non-disabled state or DFLE, i.e., DFLE = $e_x^1 + e_x^2$. For example, the DFLE for those aged 65 years was 12.5 +3.4 = 15.9 years.

Persons who did not have an activity limitation had the highest life expectancy, ALE, and DFLE, followed by those who had mild limitations. Persons who had a disability had the lowest life expectancy, ALE, and DFLE. Also, persons who had a disability had the greatest expected number of years with a disability of the three groups. For example, life expectancy at age 65 for participants with no limitation, mild limitation, and disability was 19.9, 18.6, and 17.1 years, respectively; ALE was 14.0, 9.5, and 7.2 years, respectively; DFLE was 17.3 (= 14.0+3.3), 15.2 (= 9.5+5.7), and 11.4 (= 7.2+4.2) years, respectively; and expected number of years with a disability was 2.6, 3.4, and 5.7 years, respectively. As an example, having a mild limitation at age 65 was associated with a 1.2-year (or 6%) decrease in life expectancy, 4.4-year (or 32%) decrease in ALE, 2.1-year (or 12%) decrease in DFLE, and 0.8-year (31%) increase in expected number of years with a disability. Similarly, having a disability at age 65 was associated with a 2.7-year (or 14%) decrease in life expectancy, 6.8-year (or 49%) decrease in ALE, and 5.9-year (or 34%) decrease in DFLE, and 3.1-year (120%) increase in expected number of years with a disability. All estimates are reliable with small standard errors (S2 Table). Similar results also were observed for both men and women (Table 5).

## Using a 5-level disability measure

For a 5-level disability measure, the total life expectancy, ALE, and DFLE by disability status are presented in Fig 1. As noted in this Figure, these measures vary according to level of disability in the expected manner.

## Discussion

Recently, Jia et al. (2019) examined the impact of ADL limitations on quality-adjusted life years (QALYs), a multi-dimensional summary measure of health that includes both morbidity and mortality, in the same HOS data [31]. The analysis demonstrated the incremental decrease in QALYs with worsening ADL limitations and illustrated that ADL statuses were more predictive of health-related quality of life (HRQOL) and mortality than chronic conditions. However, the analysis was based on the assumption that participants will remain in a given ADL status until death. Therefore, these estimates do not provide the exact QALY loss due to ADL, but, rather, the difference in QALYs between respondents with any ADL limitations throughout remaining lifetime and respondents without an ADL limitation throughout remaining lifetime.

**Table 3. Total expected life years and expected life years living with and without activity limitation overall and by each of two initial disability states for men and women.**

| Age ($x$) | Total sample | | | Initial disability status at age $x$ | | | | | | | |
|---|---|---|---|---|---|---|---|---|---|---|---|
| | | | | No limitation | | | Activity limitation | | | Difference[d] | |
| | $e_x$ [a] | $e_x^1$ [b] | $e_x^2$ [c] | $e_x$ | $e_x^1$ | $e_x^2$ | $e_x$ | $e_x^1$ | $e_x^2$ | total | active |
| Men | | | | | | | | | | | |
| 65 | 17.9 | 12.1 | 5.8 | 18.6 | 13.5 | 5.0 | 16.3 | 8.4 | 7.8 | 2.3 | 5.1 |
| 67 | 16.5 | 11.0 | 5.5 | 17.1 | 12.3 | 4.8 | 14.8 | 7.4 | 7.4 | 2.3 | 4.9 |
| 69 | 15.1 | 9.9 | 5.1 | 15.7 | 11.1 | 4.5 | 13.4 | 6.4 | 7.0 | 2.3 | 4.7 |
| 71 | 13.7 | 8.9 | 4.9 | 14.3 | 10.0 | 4.3 | 12.0 | 5.5 | 6.6 | 2.2 | 4.6 |
| 73 | 12.4 | 7.8 | 4.6 | 13.0 | 9.0 | 4.0 | 10.8 | 4.6 | 6.2 | 2.2 | 4.4 |
| 75 | 11.2 | 6.8 | 4.3 | 11.7 | 8.0 | 3.8 | 9.6 | 3.7 | 5.9 | 2.1 | 4.2 |
| 77 | 10.0 | 5.9 | 4.1 | 10.6 | 7.0 | 3.5 | 8.5 | 3.0 | 5.5 | 2.0 | 4.0 |
| 79 | 8.8 | 5.0 | 3.9 | 9.4 | 6.2 | 3.3 | 7.5 | 2.4 | 5.1 | 1.9 | 3.8 |
| 81 | 7.8 | 4.2 | 3.6 | 8.4 | 5.4 | 3.0 | 6.6 | 1.9 | 4.8 | 1.8 | 3.5 |
| 83 | 6.8 | 3.5 | 3.4 | 7.5 | 4.8 | 2.7 | 5.8 | 1.5 | 4.4 | 1.6 | 3.3 |
| 85 | 5.9 | 2.8 | 3.1 | 6.6 | 4.2 | 2.4 | 5.1 | 1.2 | 3.9 | 1.5 | 3.0 |
| 87 | 5.1 | 2.3 | 2.8 | 5.8 | 3.6 | 2.1 | 4.4 | 0.9 | 3.5 | 1.3 | 2.7 |
| 89 | 4.4 | 1.9 | 2.5 | 5.1 | 3.2 | 1.9 | 3.9 | 0.7 | 3.1 | 1.2 | 2.5 |
| 91 | 3.8 | 1.6 | 2.3 | 4.4 | 2.8 | 1.6 | 3.4 | 0.6 | 2.8 | 1.1 | 2.2 |
| 93 | 3.3 | 1.3 | 2.0 | 3.9 | 2.5 | 1.4 | 2.9 | 0.5 | 2.4 | 0.9 | 2.0 |
| 95 | 2.9 | 1.1 | 1.8 | 3.4 | 2.2 | 1.2 | 2.6 | 0.4 | 2.1 | 0.8 | 1.8 |
| Women | | | | | | | | | | | |
| 65 | 20.5 | 12.9 | 7.5 | 21.0 | 14.4 | 6.6 | 19.1 | 9.3 | 9.8 | 1.9 | 5.1 |
| 67 | 18.9 | 11.8 | 7.1 | 19.4 | 13.1 | 6.3 | 17.5 | 8.1 | 9.3 | 1.9 | 5.0 |
| 69 | 17.3 | 10.6 | 6.7 | 17.8 | 11.9 | 5.9 | 15.9 | 7.0 | 8.8 | 2.0 | 4.9 |
| 71 | 15.8 | 9.4 | 6.4 | 16.4 | 10.8 | 5.6 | 14.4 | 6.0 | 8.3 | 2.0 | 4.8 |
| 73 | 14.3 | 8.3 | 6.0 | 14.9 | 9.7 | 5.2 | 12.9 | 5.1 | 7.8 | 2.0 | 4.6 |
| 75 | 12.9 | 7.3 | 5.6 | 13.6 | 8.7 | 4.9 | 11.5 | 4.2 | 7.3 | 2.0 | 4.5 |
| 77 | 11.6 | 6.3 | 5.3 | 12.3 | 7.8 | 4.5 | 10.2 | 3.4 | 6.8 | 2.0 | 4.3 |
| 79 | 10.3 | 5.4 | 4.9 | 11.1 | 6.9 | 4.2 | 9.0 | 2.7 | 6.3 | 2.0 | 4.2 |
| 81 | 9.1 | 4.5 | 4.6 | 9.9 | 6.1 | 3.8 | 7.9 | 2.1 | 5.8 | 2.0 | 4.0 |
| 83 | 8.0 | 3.8 | 4.2 | 8.8 | 5.4 | 3.4 | 6.9 | 1.6 | 5.2 | 1.9 | 3.7 |
| 85 | 7.0 | 3.1 | 3.8 | 7.8 | 4.7 | 3.1 | 5.9 | 1.2 | 4.7 | 1.9 | 3.5 |
| 87 | 6.0 | 2.5 | 3.5 | 6.9 | 4.1 | 2.8 | 5.1 | 0.9 | 4.2 | 1.8 | 3.2 |
| 89 | 5.2 | 2.0 | 3.2 | 6.1 | 3.6 | 2.5 | 4.4 | 0.6 | 3.8 | 1.7 | 3.0 |
| 91 | 4.5 | 1.6 | 2.9 | 5.4 | 3.1 | 2.2 | 3.8 | 0.4 | 3.4 | 1.5 | 2.7 |
| 93 | 3.9 | 1.2 | 2.6 | 4.7 | 2.7 | 2.0 | 3.3 | 0.3 | 3.0 | 1.4 | 2.4 |
| 95 | 3.3 | 1.0 | 2.4 | 4.1 | 2.4 | 1.7 | 2.9 | 0.2 | 2.7 | 1.2 | 2.2 |

[a]: Total remaining life years (i.e., life expectancy) for persons of age $x$.

[b]: Remaining life years with no limitation (i.e., active life expectancy) for persons of age $x$.

[c]: Remaining life years with activity limitation for persons of age $x$.

[d]: difference in total life expectancy (total) and active life expectancy (active) between those without and with activity limitation; all differences are significantly different from 0 (p<0.0001).

Our study adds to the literature by providing estimates of life expectancy as well as ALE and DFLE for persons by their disability status for the U.S. community-dwelling elderly population. Use of multi-state models enables an examination of multiple and recurrent events

**Table 4. Total expected life years and expected life years in three different disability states overall and according to each of three initial disability states for U.S. older adults.**

| Age ($x$) | Total sample | | | | Initial disability status at age $x$ | | | | | | | | | | | |
|---|---|---|---|---|---|---|---|---|---|---|---|---|---|---|---|---|
| | | | | | No limitation | | | | Mild limitation | | | | Disability | | | |
| | $e_x$ [a] | $e_x^1$ [b] | $e_x^2$ [c] | $e_x^3$ [d] | $e_x$ | $e_x^1$ | $e_x^2$ | $e_x^3$ | $e_x$ | $e_x^1$ | $e_x^2$ | $e_x^3$ | $e_x$ | $e_x^1$ | $e_x^2$ | $e_x^3$ |
| 65 | 19.3 | 12.5 | 3.4 | 3.4 | 19.9 | 14.0 | 3.3 | 2.6 | 18.6 | 9.5 | 5.7 | 3.4 | 17.1 | 7.2 | 4.2 | 5.7 |
| 67 | 17.8 | 11.4 | 3.2 | 3.1 | 18.3 | 12.7 | 3.1 | 2.5 | 17.0 | 8.4 | 5.4 | 3.2 | 15.6 | 6.3 | 3.8 | 5.4 |
| 69 | 16.3 | 10.3 | 3.1 | 3.0 | 16.9 | 11.6 | 2.9 | 2.4 | 15.5 | 7.3 | 5.2 | 3.1 | 14.0 | 5.4 | 3.5 | 5.2 |
| 71 | 14.9 | 9.2 | 2.9 | 2.8 | 15.4 | 10.4 | 2.8 | 2.3 | 14.1 | 6.3 | 4.9 | 2.9 | 12.6 | 4.6 | 3.1 | 4.9 |
| 73 | 13.5 | 8.1 | 2.7 | 2.7 | 14.1 | 9.4 | 2.6 | 2.1 | 12.8 | 5.3 | 4.7 | 2.8 | 11.2 | 3.9 | 2.7 | 4.6 |
| 75 | 12.2 | 7.1 | 2.5 | 2.6 | 12.8 | 8.4 | 2.4 | 2.0 | 11.5 | 4.4 | 4.4 | 2.6 | 9.9 | 3.2 | 2.4 | 4.4 |
| 77 | 10.9 | 6.1 | 2.4 | 2.4 | 11.6 | 7.4 | 2.2 | 1.9 | 10.3 | 3.7 | 4.2 | 2.5 | 8.7 | 2.5 | 2.0 | 4.1 |
| 79 | 9.7 | 5.2 | 2.2 | 2.3 | 10.4 | 6.6 | 2.0 | 1.8 | 9.3 | 3.0 | 3.9 | 2.4 | 7.6 | 2.0 | 1.7 | 3.9 |
| 81 | 8.6 | 4.4 | 2.0 | 2.2 | 9.3 | 5.8 | 1.8 | 1.7 | 8.3 | 2.4 | 3.6 | 2.2 | 6.6 | 1.5 | 1.4 | 3.6 |
| 83 | 7.5 | 3.7 | 1.8 | 2.1 | 8.3 | 5.1 | 1.6 | 1.6 | 7.4 | 2.0 | 3.4 | 2.1 | 5.7 | 1.1 | 1.1 | 3.4 |
| 85 | 6.6 | 3.0 | 1.6 | 2.0 | 7.3 | 4.5 | 1.4 | 1.5 | 6.6 | 1.6 | 3.1 | 1.9 | 4.8 | 0.8 | 0.9 | 3.1 |
| 87 | 5.7 | 2.4 | 1.4 | 1.9 | 6.5 | 3.9 | 1.2 | 1.4 | 5.9 | 1.3 | 2.8 | 1.8 | 4.2 | 0.6 | 0.7 | 2.9 |
| 89 | 5.0 | 2.0 | 1.2 | 1.8 | 5.7 | 3.4 | 1.0 | 1.3 | 5.2 | 1.0 | 2.6 | 1.6 | 3.6 | 0.4 | 0.5 | 2.7 |
| 91 | 4.3 | 1.6 | 1.0 | 1.7 | 5.1 | 3.0 | 0.8 | 1.2 | 4.6 | 0.8 | 2.3 | 1.5 | 3.1 | 0.3 | 0.4 | 2.5 |
| 93 | 3.7 | 1.3 | 0.8 | 1.7 | 4.5 | 2.7 | 0.7 | 1.1 | 4.1 | 0.7 | 2.1 | 1.3 | 2.8 | 0.2 | 0.3 | 2.3 |
| 95 | 3.2 | 1.0 | 0.6 | 1.6 | 3.9 | 2.4 | 0.5 | 1.0 | 3.6 | 0.6 | 1.8 | 1.2 | 2.5 | 0.1 | 0.2 | 2.2 |

[a]: Total remaining life years (i.e., life expectancy) for persons of age $x$.

[b]: Remaining life years with no limitation (i.e., active life expectancy) for persons of age $x$.

[c]: Remaining life years with mild limitation for persons of age $x$.

[d]: Remaining life years with disability for persons of age $x$.

Standard errors of estimates are available in S2 Table.

simultaneously. Because of the high data requirements (i.e., longitudinal data) of the multi-state modeling method, this method has not been widely used to conduct such analyses in a large national representative sample of the U.S. elderly population [15, 24]. Among studies that did, almost all used data collected many decades ago [13–15]. The Medicare HOS is the largest longitudinal survey of the U.S. elderly population, and this data set has never been used for such an analysis. The large sample size of the Medicare HOS enabled us to examine the impact of disability on life expectancy and ALE/DFLE for older U.S. adults with good reliability (See S1 and S2 Tables). Given that chronic diseases may directly or indirectly affect a participant's disability status [32, 33], the multi-state models used in this study also can be applied for the purposes of investigating such relationships by examining transitions among a spectrum of health, ranging from healthy, to at risk, to chronic illness without impairment, to impairment, to functional limitations, to disability, and to death [33]. Additionally, we examined disability for the 5-level ADL/disability staging measure which had not been examined in the past [27].

This study addressed some analytical issues for the HOS data. First, transition times except for the date of death were interval censored and time intervals between baseline and follow-up surveys varied. We used log-linear models to estimate transition probabilities between different disability states by assuming a constant instantaneous transition rate during an age interval (i.e., piecewise-constant). To evaluate the impact of this assumption, we applied a survival model for interval-censored data that did not rely on these assumptions [30]. This method

**Table 5. Total expected life years and expected life years in three different disability states overall and according to each of three initial disability states for men and women.**

| Age | Total sample | | | | Initial disability status at age $x$ | | | | | | | | | | | |
|---|---|---|---|---|---|---|---|---|---|---|---|---|---|---|---|---|
| | | | | | No limitation | | | | Mild limitation | | | | Disability | | | |
| $(x)$ | $e_x$ [a] | $e_x^1$ [b] | $e_x^2$ [c] | $e_x^3$ [d] | $e_x$ | $e_x^1$ | $e_x^2$ | $e_x^3$ | $e_x$ | $e_x^1$ | $e_x^2$ | $e_x^3$ | $e_x$ | $e_x^1$ | $e_x^2$ | $e_x^3$ |
| | | | | | | | | Men | | | | | | | | |
| 65 | 17.9 | 12.1 | 2.9 | 2.9 | 18.6 | 13.5 | 2.8 | 2.2 | 17.1 | 9.1 | 5.1 | 2.9 | 15.6 | 7.5 | 3.0 | 5.2 |
| 67 | 16.5 | 11.0 | 2.8 | 2.7 | 17.1 | 12.3 | 2.7 | 2.1 | 15.6 | 8.0 | 4.9 | 2.7 | 14.1 | 6.5 | 2.8 | 4.8 |
| 69 | 15.1 | 9.9 | 2.6 | 2.5 | 15.7 | 11.1 | 2.5 | 2.0 | 14.2 | 7.0 | 4.7 | 2.5 | 12.7 | 5.6 | 2.6 | 4.5 |
| 71 | 13.7 | 8.9 | 2.5 | 2.3 | 14.3 | 10.0 | 2.4 | 1.9 | 12.8 | 6.0 | 4.5 | 2.4 | 11.3 | 4.7 | 2.4 | 4.3 |
| 73 | 12.4 | 7.8 | 2.4 | 2.2 | 13.0 | 9.0 | 2.2 | 1.8 | 11.6 | 5.0 | 4.3 | 2.3 | 10.0 | 3.9 | 2.2 | 4.0 |
| 75 | 11.2 | 6.8 | 2.2 | 2.1 | 11.7 | 8.0 | 2.0 | 1.7 | 10.4 | 4.2 | 4.1 | 2.2 | 8.9 | 3.1 | 1.9 | 3.8 |
| 77 | 10.0 | 5.9 | 2.1 | 2.0 | 10.6 | 7.0 | 1.9 | 1.6 | 9.3 | 3.4 | 3.9 | 2.0 | 7.8 | 2.5 | 1.7 | 3.6 |
| 79 | 8.8 | 5.0 | 1.9 | 1.9 | 9.4 | 6.2 | 1.7 | 1.5 | 8.4 | 2.8 | 3.7 | 1.9 | 6.7 | 1.9 | 1.5 | 3.4 |
| 81 | 7.8 | 4.2 | 1.8 | 1.8 | 8.4 | 5.4 | 1.5 | 1.4 | 7.5 | 2.2 | 3.4 | 1.8 | 5.8 | 1.5 | 1.2 | 3.2 |
| 83 | 6.8 | 3.5 | 1.6 | 1.7 | 7.5 | 4.8 | 1.4 | 1.3 | 6.7 | 1.8 | 3.2 | 1.7 | 5.0 | 1.1 | 1.0 | 3.0 |
| 85 | 5.9 | 2.8 | 1.4 | 1.7 | 6.6 | 4.2 | 1.2 | 1.2 | 5.9 | 1.4 | 2.9 | 1.6 | 4.3 | 0.8 | 0.8 | 2.7 |
| 87 | 5.1 | 2.3 | 1.2 | 1.6 | 5.8 | 3.6 | 1.0 | 1.1 | 5.2 | 1.2 | 2.7 | 1.4 | 3.7 | 0.6 | 0.6 | 2.5 |
| 89 | 4.4 | 1.9 | 1.1 | 1.5 | 5.1 | 3.2 | 0.8 | 1.0 | 4.7 | 1.0 | 2.4 | 1.3 | 3.2 | 0.5 | 0.4 | 2.3 |
| 91 | 3.8 | 1.6 | 0.9 | 1.4 | 4.4 | 2.8 | 0.7 | 0.9 | 4.1 | 0.8 | 2.1 | 1.2 | 2.8 | 0.3 | 0.3 | 2.1 |
| 93 | 3.3 | 1.3 | 0.7 | 1.3 | 3.9 | 2.5 | 0.5 | 0.9 | 3.6 | 0.7 | 1.9 | 1.0 | 2.5 | 0.3 | 0.2 | 2.0 |
| 95 | 2.9 | 1.1 | 0.5 | 1.3 | 3.4 | 2.2 | 0.4 | 0.8 | 3.2 | 0.7 | 1.6 | 0.9 | 2.2 | 0.2 | 0.2 | 1.9 |
| | | | | | | | | Women | | | | | | | | |
| 65 | 20.5 | 12.9 | 3.8 | 3.8 | 21.0 | 14.4 | 3.0 | 3.0 | 19.9 | 9.9 | 6.1 | 3.8 | 18.5 | 8.2 | 4.0 | 6.2 |
| 67 | 18.9 | 11.8 | 3.6 | 3.5 | 19.4 | 13.1 | 2.9 | 2.9 | 18.2 | 8.7 | 5.9 | 3.6 | 16.8 | 7.2 | 3.7 | 6.0 |
| 69 | 17.3 | 10.6 | 3.4 | 3.3 | 17.8 | 11.9 | 2.8 | 2.7 | 16.6 | 7.6 | 5.6 | 3.5 | 15.2 | 6.1 | 3.4 | 5.7 |
| 71 | 15.8 | 9.4 | 3.2 | 3.2 | 16.4 | 10.8 | 2.7 | 2.6 | 15.1 | 6.5 | 5.3 | 3.3 | 13.6 | 5.2 | 3.1 | 5.4 |
| 73 | 14.3 | 8.3 | 3.0 | 3.0 | 14.9 | 9.7 | 2.7 | 2.5 | 13.7 | 5.5 | 5.0 | 3.1 | 12.1 | 4.3 | 2.8 | 5.1 |
| 75 | 12.9 | 7.3 | 2.8 | 2.9 | 13.6 | 8.7 | 2.6 | 2.3 | 12.3 | 4.7 | 4.7 | 3.0 | 10.7 | 3.5 | 2.4 | 4.8 |
| 77 | 11.6 | 6.3 | 2.6 | 2.7 | 12.3 | 7.8 | 2.5 | 2.2 | 11.0 | 3.9 | 4.4 | 2.8 | 9.4 | 2.8 | 2.1 | 4.5 |
| 79 | 10.3 | 5.4 | 2.3 | 2.6 | 11.1 | 6.9 | 2.5 | 2.1 | 9.9 | 3.2 | 4.1 | 2.6 | 8.1 | 2.1 | 1.8 | 4.2 |
| 81 | 9.1 | 4.5 | 2.1 | 2.5 | 9.9 | 6.1 | 2.4 | 1.9 | 8.8 | 2.6 | 3.8 | 2.4 | 7.0 | 1.6 | 1.5 | 3.9 |
| 83 | 8.0 | 3.8 | 1.9 | 2.3 | 8.8 | 5.4 | 2.3 | 1.8 | 7.8 | 2.1 | 3.5 | 2.3 | 6.0 | 1.2 | 1.2 | 3.6 |
| 85 | 7.0 | 3.1 | 1.6 | 2.2 | 7.8 | 4.7 | 2.3 | 1.7 | 7.0 | 1.7 | 3.2 | 2.1 | 5.1 | 0.8 | 0.9 | 3.3 |
| 87 | 6.0 | 2.5 | 1.4 | 2.1 | 6.9 | 4.1 | 2.2 | 1.5 | 6.2 | 1.3 | 2.9 | 1.9 | 4.3 | 0.6 | 0.7 | 3.1 |
| 89 | 5.2 | 2.0 | 1.2 | 2.0 | 6.1 | 3.6 | 2.1 | 1.4 | 5.5 | 1.0 | 2.7 | 1.8 | 3.7 | 0.4 | 0.5 | 2.8 |
| 91 | 4.5 | 1.6 | 1.0 | 1.9 | 5.4 | 3.1 | 2.0 | 1.3 | 4.8 | 0.8 | 2.4 | 1.6 | 3.2 | 0.3 | 0.4 | 2.6 |
| 93 | 3.9 | 1.2 | 0.9 | 1.8 | 4.7 | 2.7 | 2.0 | 1.2 | 4.3 | 0.6 | 2.2 | 1.4 | 2.8 | 0.2 | 0.2 | 2.4 |
| 95 | 3.3 | 1.0 | 0.7 | 1.7 | 4.1 | 2.4 | 1.9 | 1.1 | 3.8 | 0.5 | 2.0 | 1.3 | 2.5 | 0.1 | 0.2 | 2.3 |

[a]: Total remaining life years (i.e., life expectancy) for persons of age $x$.

[b]: Remaining life years with no limitation (i.e., active life expectancy) for persons of age $x$.

[c]: Remaining life years with mild limitation for persons of age $x$.

[d]: Remaining life years with disability for persons of age $x$.

provided similar estimates. The difference of ALE/DFLE estimates at age 65 between these two methods was ≤0.1 years. However, not all interval-censored survival models had a solution and, when there was a solution, its estimates were not as reliable as estimates based on the log-linear model.

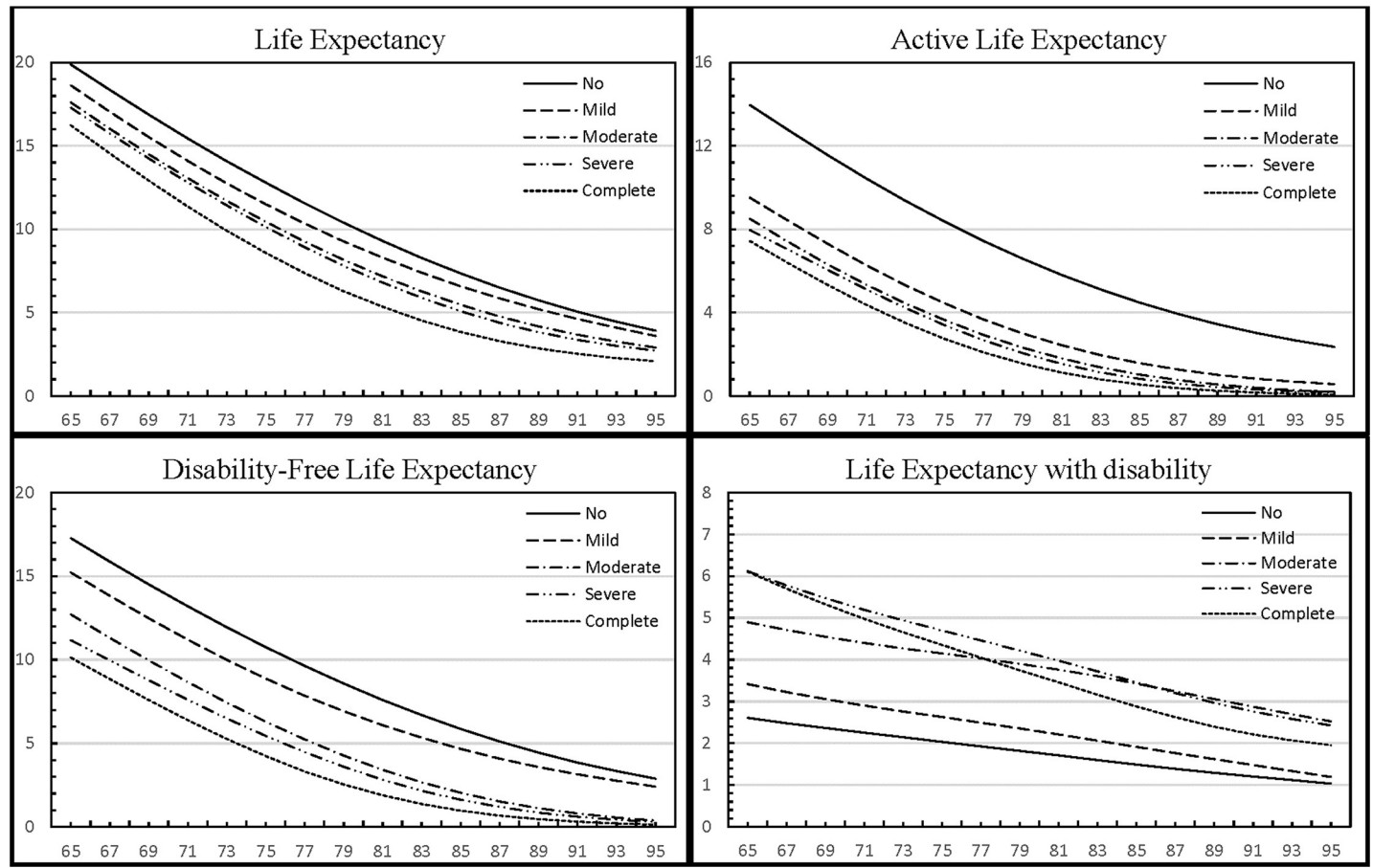

**Fig 1. Life expectancy, active life expectancy, disability-free life expectancy, and life expectancy with disability by five disability statuses at different ages among older U.S. adults.** No: Stage 0, no difficulty; Mild: Stage I, mild limitation; Moderate: Stage II, moderate limitation; Severe: Stage III, severe limitation; Complete: Stage IV, complete limitation.

Second, probabilities of death estimated from the HOS were unreliable due to the short follow up time. Furthermore, estimates might be biased because the HOS excluded institutionalized persons, and persons in poor health might be less likely to participate. We used a method that assumed that the HOS samples had the same age-specific mortality rates as the U.S. general population to improve reliability and validity of estimates. This is because the HOS data may be used to monitor the health of the elderly general population [34]. To validate this assumption, we used a parametric (Weibull) survival model to estimate probabilities of death in two years from the HOS data. The estimated life expectancy with the survival model was nearly the same as that of the U.S. life table (the difference was only 0.01 years). This also demonstrated the validity of using the HOS data to estimate life expectancy for the U.S. population aged 65 and older.

This study has some limitations. First, because this analysis used data from the HOS, a survey of Medicare beneficiaries who voluntarily enrolled in private Medicare Advantage health plans, the sample may be younger and healthier than the overall Medicare population [35]. However, our analysis showed that life expectancy estimated based on the HOS samples was nearly the same as the life expectancy for the U.S. general population. Second, potential bias might exist due to lack of participation in the follow-up survey as, for example, respondents now might be institutionalized. However, there was no difference in baseline characteristics,

including age, sex, race, disability status, and chronic conditions, between those who participated and those who did not participate in the follow-up survey. Third, respondents reported their own limitations in their ADLs, which were not validated by medical examinations. Certain ADL items may have a wider interpretation due to such factors as culture, education, and language. Fourth, the first-order Markov process assumes the same transition probabilities for all persons of a similar age. Previous disability status and length of time having a disability may impact transition probabilities. Fifth, we assumed only a single transition from baseline to follow-up. This assumption might lead to underestimating the impact of disability on ALE and DFLE [14, 36]. However, some investigators argued that the impact of this assumption on ALE/DFLE estimates was relatively small [15, 37].

## Conclusions

This study utilized a large legacy data set of the U.S. elderly population, the Medicare HOS, to conduct a multi-state modeling analysis of complex transitions among disability states. This study highlights how reporting greater levels of disability adversely impacts life expectancy and active life expectancy among older persons at all ages examined. These analyses might enable practitioners to not only estimate what percentage of life years an elderly person would be expected to spend in an independent state, but also how delayed onset and speedy recovery of disability would increase life expectancy. Such assessments could have implications for future health care planning. In terms of measuring disability, this study generates public health data that will provide evidence on projecting disability among a growing elderly population. Ultimately, this holistic model will be of assistance in more accurately forecasting future health care needs, as well as projected expenditures, in the United States elderly population.

## Supporting information

**S1 Table. Standard error of estimates in Table 2.** [a]: Standard error (SE) of estimated life expectancy at age $x$. [b]: SE of estimated active life expectancy at age $x$. [c]: SE of estimated life expectancy with activity limitation at age $x$. [d]: SE of estimate difference in total life expectancy (total) and active life expectancy (active).
(PDF)

**S2 Table. Standard error of estimates in Table 4.** [a]: Standard error (SE) of estimated life expectancy at age $x$. [b]: SE of estimated active life expectancy at age $x$. [c]: SE of estimated life expectancy with mild limitation at age $x$. [d]: SE of estimated life expectancy with disability at age $x$.
(PDF)

## Author Contributions

**Conceptualization:** Haomiao Jia, Erica I. Lubetkin.

**Data curation:** Haomiao Jia.

**Formal analysis:** Haomiao Jia.

**Investigation:** Haomiao Jia.

**Methodology:** Haomiao Jia.

**Project administration:** Haomiao Jia.

**Resources:** Haomiao Jia.

**Software:** Haomiao Jia.

**Supervision:** Haomiao Jia.

**Validation:** Haomiao Jia.

**Visualization:** Haomiao Jia.

**Writing – original draft:** Haomiao Jia, Erica I. Lubetkin.

**Writing – review & editing:** Haomiao Jia, Erica I. Lubetkin.

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
