## [Decision Letter · Decision Letter 0]

10 Jul 2020

PONE-D-20-17729

Life expectancy and active life expectancy by disability status in older U.S. adults: results from the Medicare Health Outcome Survey (HOS)

PLOS ONE

Dear Dr. Jia,

Thank you for submitting your manuscript to PLOS ONE. After careful consideration, we feel that it has merit but does not fully meet PLOS ONE’s publication criteria as it currently stands. Therefore, we invite you to submit a revised version of the manuscript that addresses the points raised during the review process. Please submit your revised manuscript by July 30, 2020. If you will need more time than this to complete your revisions, please reply to this message or contact the journal office at plosone@plos.org. Please include the following items when submitting your revised manuscript:

We look forward to receiving your revised manuscript.

Kind regards,

Hemachandra Reddy

Academic Editor

PLOS ONE

Journal Requirements:

2. In ethics statement in the manuscript and in the online submission form, please provide additional information about the patient records used in your retrospective study. Specifically, please ensure that you have discussed whether all data were fully anonymized before you accessed them and/or whether the IRB or ethics committee waived the requirement for informed consent. If patients provided informed written consent to have data from their medical records used in research, please include this information.

3. Please consider including your recent study (Jia, Haomiao, et al. "Quality-adjusted life years (QALYs) associated with limitations in activities of daily living (ADL) in a large longitudinal sample of the US community-dwelling older population." Disability and health journal 12.4 (2019): 699-705.) in the discussion, to provide a more complete review of the topic.

Reviewers' comments:

Reviewer's Responses to Questions

**Comments to the Author**

1. Is the manuscript technically sound, and do the data support the conclusions?

Reviewer #1: Yes

Reviewer #2: Yes

2. Has the statistical analysis been performed appropriately and rigorously? 

Reviewer #1: I Don't Know

Reviewer #2: I Don't Know

3. Have the authors made all data underlying the findings in their manuscript fully available?

Reviewer #1: Yes

Reviewer #2: Yes

4. Is the manuscript presented in an intelligible fashion and written in standard English?

Reviewer #1: Yes

Reviewer #2: Yes

5. Review Comments to the Author

Reviewer #1: I'm not really picking up on any existential flaws, although a statistician might be better at critically probing the modeling approach. It's basically a secondary analysis of a large data set that shows health status relates to both active and disabled lifespans, which seems unsurprising. It's certainly well-written and I suspect there are some in the public health field who might find this of interest.

Reviewer #2: Upon reviewing the manuscript , I felt that it was well drafted and scientifically coherent. It is a novel topic as it explores the unique question of how many years of life expectancy years are spent active and independent vs in disability requiring assistance. One recommendation I have is for the author is elucidate upon the chronic disorders that the participants had and how those directly or indirectly affected their disability status. As the study aims to create a database for healthcare providers, a sound understanding of the various chronic conditions like diabetes, hypertension, obesity, etc of the participants and how those posed as a disability in their lives would help healthcare providers predict life expectancy and active life expectancy for those who have those specific conditions. Special mention could also be made of disorders that cause cognitive decline like Alzheimer's Disease and dementia and how those too can impact the disability status and life expectancy of a person. This additional knowledge can help health care providers design specific treatment plans catered to the specific disorders and help them increase the life expectancy and/or the active life expectancy of their geriatric patients.

6. PLOS authors have the option to publish the peer review history of their article (what does this mean?). If published, this will include your full peer review and any attached files.

Reviewer #1: No

Reviewer #2: No

---

## [Author Response · Author response to Decision Letter 0]

18 Aug 2020

Editor comments and journal Requirements:

Response: We have formatted the manuscript to meet PLOS ONE’s requirements 

2. In ethics statement in the manuscript and in the online submission form, please provide additional information about the patient records used in your retrospective study. Specifically, please ensure that you have discussed whether all data were fully anonymized before you accessed them and/or whether the IRB or ethics committee waived the requirement for informed consent. If patients provided informed written consent to have data from their medical records used in research, please include this information. 

Response: This study is a secondary data analysis using the Limited Data Set (LDS) of the HOS from the CMS. The HOS LDS files retain some protected beneficiary-level health information, such as date of birth, gender, race/ethnicity, and county of residence; however, specific direct person identifiers (i.e., name, address, HIC number, and SSN) are not included in the LDS files, as outlined in the Health Insurance Portability and Accountability Act (HIPAA Privacy Rule) (https://www.hosonline.org/en/data-dissemination/research-data-files/).

3. Please consider including your recent study (Jia, Haomiao, et al. "Quality-adjusted life years (QALYs) associated with limitations in activities of daily living (ADL) in a large longitudinal sample of the US community-dwelling older population." Disability and health journal 12.4 (2019): 699-705.) in the discussion, to provide a more complete review of the topic.

Response: We added the following to the Discussions: 

Recently, Jia et al. (2019) examined the impact of ADL limitations on quality-adjusted life years (QALYs), a multi-dimensional summary measure of health that includes both morbidity and mortality, in the same HOS data [31]. The analysis demonstrated the incremental decrease in QALYs with worsening ADL limitations and illustrated that ADL statuses were more predictive of health-related quality of life (HRQOL) and mortality than chronic conditions. However, the analysis was based on the assumption that participants will remain in a given ADL status until death. Therefore, these estimates do not provide the exact QALY loss due to ADL, but, rather, the difference in QALYs between respondents with any ADL limitations throughout remaining lifetime and respondents without an ADL limitation throughout remaining lifetime. 

Response: The data used in this study were from a LDS from CMS. The dataset contains potentially identifying or sensitive patient information (e.g., participants’ zip code, date of birth, date of death, etc.). A signed Data Use Agreement (DUA) with CMS is required to obtain LDS data files. The data file was obtained through a DUA between the U.S. Center for Disease Control and Prevention (CDC) and the CMS. In this DUA, the corresponding author (HJ) is one of custodians of the data file. The authors do not have the authority share the data publically. 

Reviewer comments:

Reviewer's Responses to Questions

Comments to the Author

1. Is the manuscript technically sound, and do the data support the conclusions?

Reviewer #1: Yes

Reviewer #2: Yes

2. Has the statistical analysis been performed appropriately and rigorously? 

Reviewer #1: I Don't Know

Reviewer #2: I Don't Know

3. Have the authors made all data underlying the findings in their manuscript fully available?

Reviewer #1: Yes

Reviewer #2: Yes

4. Is the manuscript presented in an intelligible fashion and written in standard English?

Reviewer #1: Yes

Reviewer #2: Yes

5. Review Comments to the Author

Reviewer #1: I'm not really picking up on any existential flaws, although a statistician might be better at critically probing the modeling approach. It's basically a secondary analysis of a large data set that shows health status relates to both active and disabled lifespans, which seems unsurprising. It's certainly well-written and I suspect there are some in the public health field who might find this of interest.

Reviewer #2: Upon reviewing the manuscript, I felt that it was well drafted and scientifically coherent. It is a novel topic as it explores the unique question of how many years of life expectancy years are spent active and independent vs in disability requiring assistance. One recommendation I have is for the author is elucidate upon the chronic disorders that the participants had and how those directly or indirectly affected their disability status. As the study aims to create a database for healthcare providers, a sound understanding of the various chronic conditions like diabetes, hypertension, obesity, etc of the participants and how those posed as a disability in their lives would help healthcare providers predict life expectancy and active life expectancy for those who have those specific conditions. Special mention could also be made of disorders that cause cognitive decline like Alzheimer's Disease and dementia and how those too can impact the disability status and life expectancy of a person. This additional knowledge can help health care providers design specific treatment plans catered to the specific disorders and help them increase the life expectancy and/or the active life expectancy of their geriatric patients.

Response: We agree with the reviewer that chronic disorders may directly or indirectly affect the participant’s disability status. Other investigators and organizations have examined the intersection between chronic diseases and disability (Gulley et al, 2018). For example, the Institute of Medicine’s Living well with chronic illness: A call for public health action describes a distribution of population level subgroups that spans from healthy, to at risk, to chronic illness without impairment, to impairment, to functional limitations, and to disability (IOM 2012). We previously published a manuscript the impact of ADLs on quality-adjusted life years (QALYs) (Jia et al. 2019). The analysis illustrated that ADL statuses were more predictive of health-related quality of life (HRQOL) and mortality than chronic conditions. We have added a paragraph on the findings of this manuscript, and how the current investigation adds to the literature, but a full discussion of the impact of chronic diseases is beyond the scope of this manuscript.

We also agree with the reviewer that disorders associated with cognitive decline impact the disability status and life expectancy of a person. Unfortunately, the Medicare HOS Cohort 15 did not include an item about memory problems. However, the more recent versions of the HOS (2018, 2019, and 2020) include the following item: In the past month, how often did memory problems interfere with your daily activities? This would be an interesting item to examine in a future analysis. Of note, CMS has not announced the date of dissemination of HOS data from 2018 or later.

References: 

• Jia H, Lubetkin EI, DeMichele K, Stark DS, Zack MM, Thompson WW. Quality-adjusted life years (QALYs) associated with limitations in activities of daily living (ADL) in a large longitudinal sample of the U.S. community-dwelling older population. Disabil Health J. 2019; 12(4):699-705.

• Gulley SP, Rasch EK, Bethell CD, Carle AC, Druss BG, Houtrow AJ, Reichard AJ, Chan L. At the intersection of chronic disease, disability and health services research: A scoping literature review. Disabil Health J. 2018; 11(2): 192–203.

• Institute of Medicine. Living well with chronic illness: A call for public health action. Washington, DC: The National Academies Press; 2012.

6. PLOS authors have the option to publish the peer review history of their article (what does this mean?). If published, this will include your full peer review and any attached files.

Do you want your identity to be public for this peer review? For information about this choice, including consent withdrawal, please see our Privacy Policy.

Reviewer #1: No

Reviewer #2: No

---

## [Editor Report · Decision Letter 1]

26 Aug 2020

Life expectancy and active life expectancy by disability status in older U.S. adults: results from the Medicare Health Outcome Survey (HOS)

PONE-D-20-17729R1

Dear Dr. Jia

We’re pleased to inform you that your manuscript has been judged scientifically suitable for publication and will be formally accepted for publication once it meets all outstanding technical requirements.

Kind regards,

Hemachandra Reddy

Academic Editor

PLOS ONE
---

## [Editor Report · Acceptance letter]

15 Sep 2020

PONE-D-20-17729R1 

Life expectancy and active life expectancy by disability status in older U.S. adults: results from the Medicare Health Outcome Survey (HOS) 

Dear Dr. Jia:

I'm pleased to inform you that your manuscript has been deemed suitable for publication in PLOS ONE. Congratulations! Your manuscript is now with our production department. 

Kind regards, 

on behalf of

Dr. Hemachandra Reddy 

Academic Editor

PLOS ONE